# Numerical and Experimental Study on the Steel Strands under the Coupling Effect of a Salt Spray Environment and Cyclic Loads

**DOI:** 10.3390/ma13030736

**Published:** 2020-02-06

**Authors:** Xuanrui Yu, Guowen Yao, Lifeng Gu, Weiqing Fan

**Affiliations:** 1School of Civil Engineering, Chongqing Jiao Tong University, Chong Qing 400074, China; 2State Key Laboratory of Mountain Bridge and Tunnel Engineering, Chong Qing 400074, China

**Keywords:** sochastic pitting-corrosion model, neural network method, indoor experiment, multi-dimensional linear regression method

## Abstract

Composed of multi-strand parallel high-strength wires or steel strands, the stayed cables have been widely used recently in stayed bridges or suspension bridges owing to their light weight and high bearing capacity, especially the steel strands. Meanwhile, chloride-induced corrosion of steel strands is one of the most considerable factors for the durability of the stayed cable exposed to marine environments. The fatigue caused by both cyclic loading and corrosion can affect the life of the steel strands. Besides, the current studies related to the effects of the aforementioned two impact factors on the life of the steel strands either considered the fatigue only, or took the two impact factors into account separately. The coupling effects of fatigue and corrosion on the life of the steel strands are required to be further explored and discussed. Consequently, it is essential to create a model to predict the life of the steel strands with the coupling effects taken into consideration. In this paper, an indoor physical experiment of the steel strand specimens exposed to marine tidal environment was carried out. To avoid accidental errors, the whole specimens were divided into 20 groups, with each group having two steel wires with a 5 days, 10 days, and 15 days cycle for the test. The corrosion of steel strands was observed at various exposure times and it was found that the pits were formed on the surface with the chloride ion erosion to the steel strands. Deeper and sharper pits result in greater pitting-local stress and a shorter fatigue life, which is also the main reason for reducing the carrying capacity of the steel strands. However, a detailed description for this problem is lacking in current domestic and foreign literature, because the pit is hard to predict owing to its complex nature. In order to simulate the evolution of the pits, the stochastic pitting-corrosion model was set up by the neural network method to evaluate the pit evolutions over time. In addition, an empirical formula consisting of length–width ratios, length–depth ratios, and depth-to-width ratios of the pits was obtained to determine the stress concentration factor based on the multi-dimensional linear regression method. The fatigue notch factor of components can be deduced by the stress concentration factor, and the life of the steel strands can be deduced by both of them. The findings are expected to be useful in realistically predicting the durability of wire structures.

## 1. Introduction

Cable-stayed bridges and suspension bridges have become quite popular in recent years. As a considerable member of the cable-stayed bridges and suspension bridges, the durability and reliability of cables must be guaranteed. The design of stay cables must face challenges including corrosion, fatigue, and even both of these [1]. In general, the design period of the cable-stayed bridges and suspension bridges is about 100 years, but most of the actual engineering data show that the design life of stay cables only approaches 20 years, and is even lower when exposed to the marine environment [2,3]. However, few studies in the literature have put forward any method to assess the fatigue life of the parallel cables. In early stages, the Weibull distribution, often used to describe the fatigue life of wires, is supported by experimental results. The fatigue life model of wire cables can be derived by the parameterized fatigue life model of a single wire. The models were adopted by Rackwitz and Faber [4]. In order to predict the fatigue life of strand cables, Stallings and Frank [5] used Monte Carlo simulations to deduce it and additionally obtained the empirical formula. However, these tests only exhibit the fatigue life caused by the load, but not the effects caused by the corrosion mentioned.

Corrosion is a crucial factor to determine the resident life of the cables. The stay cables exposed to the environment are easily affected by acid ions, especially the sea-crossing bridges. There are a great many old suspension bridges in the United States, and some of them have suffered deteriorated cables in recent years. Steel wires in the bridge cables are heavily corroded and many of them are broken. It was found that the corrosion degree is related to the concentration of chloride ion and the humidity of the environment [6].

The chloride ions penetrate the oxide layer and the protective layer on the surface of metal materials, and then the electrochemical reaction will occur between the chloride ion and internal metal matrix. At the same time, adsorbing the hydration energy of chloride ions into the pores and cracks on the surface will squeeze or replace the oxygen in the chlorination layer, the original insoluble oxide would be displaced by soluble chloride, and the state of metal surface will change from passivation to liveliness, which leads to the steel wires being corroded in the end [7]. The main categories of the corrosion can be divided into two aspects, hydrogen absorption corrosion and oxygen absorption corrosion, both of which can apparently lead to the decrease of the fatigue life of steel strands. Because of the pits formed on the surface of the steel strands in the process of the corrosion [8], a deeper and sharper pit will result in greater pitting-local stress and a shorter fatigue life [9]. Moreover, pitting is hard to predict owing to its complex nature, which involves the type of attack corrosion [10,11], attacked material [12,13], environmental temperature, and so on [14,15], and it is rather complicated to predict the evolution of the pits. Some researchers have studied pitting by tests in the past decades, and find that the distribution forms of the maximum depth of pits are consistent with the Gumbel extreme distribution, and the width values of erosion pits obey the logarithmic normal distribution [16]. Besides, C. Casavola [17], by an indoor physical tensile test with laboratory-corroded wires, finds that the brittleness of the corroded wires is mainly caused by pitting-stress concentrations.

To study the evolution of the erosion pits, an indoor physical experiment for steel strands specimens exposed to a salt spray environment was carried out. The depth of pits at various exposure times were tested with the consideration of the pit randomness. This article used the neural network method to simulate the evolution of the pits. An empirical formula consisting of length–width ratios, length–depth ratios, and depth-to-width ratios of the pits was obtained to determine the stress concentration factor based on the multi-dimensional linear regression method simultaneously. Accordingly, the resident fatigue life of steel strands can be predicted, but also provided a reference for the further analysis of cables corrosion fatigue life.

## 2. The Test of Corroded Steel Strands

A salt spray corrosion test chamber was set up to simulate the corrosion process of steel strands, with the consideration of the combined effects of corrosion and load on steel strands. The maximum tensile stress of the alternating load is 740 MPa with the stress amplitudes of Δσ = 100 MPa, Δσ = 200 MPa, Δσ = 300 MPa, respectively. Four hours is taken as a loading cycle, and the upper limit and lower limit of the steel strand stress are switched every 2 h. Taking Δσ = 200 MPa as an example, the loading mode is shown in Figure 1. The entire test is divided into 20 groups, with two steel wires in each group tested at 5 days, 10 days, and 15 days cycles to avoid accidental errors. The length of the wire is related to the size of the test chamber. A length of 5.0 m is taken as the length of the steel wire for the convenience of observing the variation of the pit depth considering factors such as the test site comprehensively. The diameter of the steel strand used in the test is 5 mm, the tensile strength is 1860 MPa, and the weight of the galvanized layer of a single steel strand is ≥110 g/m^2^. The instruments used in the experiment are shown in Table 1.

### 2.1. Test Process and Conditions

The salt spray corrosion test chamber (Figure 2) is the primary experimental apparatus of the artificially accelerated acid rain experiment. With few literature standards provided for the artificially-accelerated acid rain experiments home and abroad, the corrosive gas used in this experiment was based on the chemical composition of normal atmospheric acid rain. The reason is that the acid ion concentration in the air is susceptible to factors such as temperature and humidity. These factors will be affected by the region and the climate, of which the randomness is too excessive to predict accurately. Perfectly simulating the corrosion state of steel strands in actual engineering is rather difficult. The concentration of the corrosive liquid used in the test is adjusted to ten times the actual situation to perform the artificial accelerated test according to the corresponding weather forecast and the combination of the reference [18,19,20] to ensure the accuracy of the test with the consideration of random factors such as climate changes. The temperature control device adopted in this experiment was based on the temperature devices used in acetic acid salt spray tests. The temperature is maintained at 25 degrees Celsius. Consisting of sodium chloride, glacial acetic acid, copper chloride dehydrate, and distilled water, the concentration of the acid solution is controlled at (50 ± 5) g/L. The pH value of the acid solution is controlled to be between 3.1 and 3.3, and the specific ingredients of the acid solution are shown in Table 2.

The salt spray corrosion test device is shown in Figure 2. The surface of the device is painted with anti-corrosive paint to prevent corrosion. While applying the load provided by the jacks on both sides of the test box and the reaction force wall, corrosive liquid is sprayed from the nozzle in the form of mist to fill the entire device, with which the steel strand is completely exposed to the salt spray environment. The results are shown in Figure 3.

The corrosion results of the steel strands in the salt spray test are shown in Figure 3. After 720 h, a large area of corrosion appeared on the surface of the steel strands, and the cross section significantly narrowed. Subsequently, the specimens are cleaned with acid solution and purified water and blow-dried with cold air. KYKY-2008B electronic scanner is adopted in measuring the size of the pits. The scan results are partially shown below.

Figure 4 depicts that a large number of pits are generated on the surface of the steel strands, and numerous cracks formed around the pits, part of which have a tendency to expand further, which may cause severe impact on the fatigue life of steel strands.

### 2.2. Test Results

The test results under three stress amplitudes are shown below.

Grayscale analysis is performed on the corrosion pictures with GSA Image Analyser software and the pictures are enlarged by 170 times [18] to get a clearer and more visible pit corrosion topography map. To calculate the pit density, select gray in color depth, measure the hue at the center and edge of the pit with the color indicator function, derive a histogram of the tone distribution, and compute the statistical distribution after defining the hue value below 110 as the pits. The standard deviation can indicate the dispersion degree of a group of data. A larger standard deviation results in a more uniform corrosion effect. With the histogram exhibiting the gray and white distribution, the vertical axis in the figure represents the occurrence frequency of a certain gray level (the number of pixels in the gray level), while the abscissa represents the tone level of 0 to 255 (0 is black, 255 is white), with which the vivid distribution of the gray level can be obtained.

Figure 5 describes that when the stress amplitude is 100 MPa, the surface of the steel strand sample is relatively flat with a small number of pits about 40 μm after 720 h of corrosion. No obvious pits, but some pitting phenomena and short shallow trenches generated by connecting the pits appear compared with the 170× image. The average pit width is between 30 μm and 50 μm, and lateral cracks with a length of about 200 μm and a width of about 20 μm formed partially with the stress amplitude of 200 MPa. Although the density of pits on the surface is great, the average depth is small under the microscope. The surrounding pits will link together with constant growth. Besides, the pits can be observed with the naked eye when the stress amplitude is 300 MPa. The high-magnification picture shows that the pits have developed into trenches whose width reached 200 μm, which is 10 times the stress amplitude at 100 MPa, while the longitudinal grooves also have a width of 50 μm.

In addition, from the histogram after grayscale processing, the grayscale histogram has the best dispersion, of which the grayscale standard deviation is the best among the three loading conditions with the stress amplitude of 100 MPa. When the stress amplitude reaches 200 MPa, the derivation speed of the pit has accelerated, and microscopic cracks appear with the connection of the pits. The loading and unloading processes accelerate the development of pits with concentrated gray tone value when the alternating stress amplitude is 300 MPa. The analysis results in various cases are shown in Table 3.

Table 3 shows that when Δσ=100 MPa, the local corrosion degree of the steel strands is 32.1%, while it is 39.4% when Δσ=200 MPa, and the highest corrosion degree is 54.79% when Δσ=300 MPa, indicating that the magnitude of the stress amplitude directly affects the corrosion degree of the steel strands. The results of the pit size measured by KYKY-2008B electronic scanner are shown in Figure 6.

## 3. Stochastic Pitting-Corrosion Modelling

Building a pit-prediction model to simulate the pit evolution is quite complicated because of the depth of the pit sites. Further, considering the randomness along the process of the pits formed, the article concludes with the derivation laws of the pit in the first 300 h and acquires 100 sets of data, which are regarded as raw time series with the observation of a single corrosion phenomenon on the surface of steel strands (observed every 3 h) in the salt spray corrosion test. Models built with the first 85 groups of data selected as the sample size through the nonlinear auto regressive (NAR) neural network are compared with the remaining 15 groups of data to judge the accuracy of the model to further predict the derivation law of the pit with accuracy.

### 3.1. Theory of Neural Network Method

Consisting of an input layer, an output layer, and a hidden layer, the NAR (nonlinear auto regressive) neural network is a non-linear autoregressive model and a time series prediction model based on a dynamic neural network. The historical data are input to the input layer, the hidden layer neurons are weighted to form the input of the output layer, and the output layer outputs the predicted value of future data. Each output in the model is directed to the input of the neural network layer, and the error value between the fitted value and the actual value is fed back to the neural network as the basis in adjusting the parameters of the next output to complete the adjustment of the neural network. The derivation of a pit is a random, non-linear changing process that is hard to describe quantitatively by traditional methods. However, the NAR neural network is a method related to time series, with which the future evolution trend can be predicted by analyzing the historical data’s changing characteristics accurately to achieve the most probable depth of the pits. At present, predicting random models changing over time and nonlinear models by the NAR neural network method is applicable.

### 3.2. The NAR Model

The learning and training steps of the NAR neural network are shown in Figure 7—the input on the left represents the input data, the output on the right represents the output data, w is the connection weight, and b denotes the threshold [21].

The fitting degree between the calculated results and the measured results is the best when m = 10, as is proved with a large amount of calculations. The fitting results calculated by setting different delay levels are shown below.

Figure 8 exhibits the fitting degree between the calculated and measured values. When m = 10, the fitting coefficient of the measured value and the simulated value is 0.98, confirming the applicability of the model. The simulation results are shown below. Figure 9 and Figure 10 exhibit the autocorrelation error and the relative error value between the simulation result and the measured value, respectively. Except when lag = 0 (zero-order autocorrelation), all the other autocorrelation coefficients do not exceed the upper and lower confidence intervals, and all the errors are also within a limited range, as shown in Figure 11, which proves again the feasibility of the NAR neural network method for the prediction of the pits. Furthermore, the prediction of the last 10 test data is shown in the red part of Figure 12, and the blue part represents the measured data; the change rules of the two are visibly close and the slope of the curve gradually decreases with the passage of time, indicating that the derivation rate of the pits gradually decreases, which is consistent with the results obtained in the work of [10] in the meantime.

## 4. The Finite Element Model of Pit

Exposed to the salt spray corrosion environment, elliptical pits of different sizes will form on the surface of the steel strands. Furthermore, strong local stress would emerge at the pit positions under the action of tensile stress, which may cause an obvious stress concentration phenomenon and even the derivation of cracks. In general, the stress concentration factor is used to depict the stress concentration levels. A larger stress concentration factor will lead to a more obvious stress concentration phenomenon, and the less obvious it is otherwise. The stress concentration factor (SCF) is the ratio of the local maximum stress to the nominal stress. Plenty of methods to calculate the stress concentration factors have been developed for elliptic holes at present [22,23], however, the objects of the above studies are concentrated on penetrating small holes, while the pits studied are usually formed on the surface of steel strands that are similar to non-penetrating oval holes. Complicated mechanics of non-penetrating elliptical holes increase the difficulty in finding its analytical solution. Meanwhile, numerical simulation analysis, as a convenient, fast, and accurate method in engineering practice, is simultaneously widely used to explore the relationship between the sizes of the pits and the stress concentration factor.

### 4.1. Conditions of the Pit Model

The length of the steel strands is 10 mm with the diameter of 5 mm. The basic parameters of the model are as follows: elastic modulus = 2 × 10^10^ MPa, elastic Poisson’s ratio = 0.3, and element type is solid45 solid element (high-order element solid95 is used for extremely irregular parts). Dividing the model with mapping grids from top to bottom to improve the calculation speed without losing its accuracy, the pit sizes are consistent with the test results. Furthermore, the number of elements and nodes of the model are 36,782 and 65,437, respectively. The grid diagram is shown in Figure 13. The schematic diagram of the calculation model is shown in Figure 14. One end of the model is the fixed constraint (shown in the red position), and the other end is uniaxially loaded. The calculation results are shown in Figure 15. The figure depicts the stress cloud diagram and the maximum stress value at the pit positions. Besides, the location of the maximum value may be at the bottom of the pit, or in the direction derived from the length or width of the pit. The specific location is related to the sizes of the pit (shown by the black arrow in Figure 15).

Calculation method of stress concentration coefficient $:\;SCF=\frac{\sigma_{i}}{\sigma_{0}}$, where $\sigma_{i}$ is the local maximum stress and $\sigma_{0}$ is the nominal stress.

### 4.2. The Simulation Result

The model is fixed at one end and loaded at the other end. The results obtained are partially shown below.

It is known from the results that a visibly tensile stress zone appears at the pit location. The maximum equivalent stress appears at the pit bottom when the pit depth is shallow (d < 0.6 mm). With the increase of the depth, the high stress areas gradually shift to the lateral edge, and subsequently, the bottom of the pits gradually become non-dangerous areas, which indicates that the areas that are prone to crack are located at the bottom of the pits when the corrosion is slight and the depth is shallow, and the easy-to-crack areas are located at the lateral edge of the pits when the corrosion is serious and the depth of the pits is deep simultaneously—this shows that the length and width of the pits have a certain impact on the local stress distribution. Pit models with different length–width ratios, length–depth ratios, and depth-to-width ratios are established to take the effects of the length, width, and depth into consideration on the local stress comprehensively. The results are partially shown in Figure 16.

In the figure, w denotes the pit width, L means the pit length, and d is the pit depth. The local maximum stress of the pits with different length–width ratios, length–depth ratios, and depth-to-width ratios can be obtained through the model, and from these, stress concentration factors could be calculated. Additionally, the length, width, and depth of the pits are directly proportional to the stress concentration factors at the pit position. Accordingly, this paper explores the connection between the stress concentration factors and the length–width ratios, length–depth ratios, and depth-to-width ratios of the pit based on the multiple linear regression method.

## 5. Multidimensional Linear Regression Theory

Depending on the idea of the probability theory, analyzing the sensitivity of each factor to the calculation result intuitively with the multidimensional linear regression theory is widely used in practical engineering. The specific expression is as follows [24]:(1)F=βΔ+ξ

In Equation (1), in order to eliminate the interference of other factors except the pit itself, we assume that ξ obeys the normal distribution $\left(0,\sigma^{2}\right)$. Each matrix is shown as follows:(2)$$\Delta =\{\begin{array}{cccc} 1 & a_{1} & b_{1} & c_{1}\\ \vdots & \vdots & \vdots & \vdots \\ 1 & a_{n} & b_{n} & c_{n} \end{array}\}\;F=\{\begin{array}{c} F_{1}\\ \vdots \\ F_{n} \end{array}\}$$
where $\xi ={\{\begin{array}{ccc} \xi_{1} & \cdots & \xi_{n} \end{array}\}}^{T}$, $\beta ={\{\begin{array}{cccc} \beta_{0} & \beta_{1} & \cdots & \beta_{2} \end{array}\}}^{T}$; n means the number of the pits; $a_{i}$, $b_{i}$, $c_{i}$ denote the length-to-width ratios, length-to-depth ratios, and width-to-depth ratios of the pits, respectively; and $F_{i}$ is the stress concentration factor corresponding to the pit.

### 5.1. Parameter Estimation

In the linear regression process, the solution to the regression coefficient β is obtained by the least-square method in most instances. The expression is as follows [24,25]:(3)$$\overset{\wedge }{\beta }={\left(\Delta^{T}\Delta \right)}^{-1}\Delta^{T}Y$$

Substituting Equation (3) into Equation (1), we get the following:(4)$$\overset{\wedge }{F}=\overset{\wedge }{\beta }\Delta$$

### 5.2. Significant Test of Regression Model

In general, two statistical indicators, including $\tilde{F}$ and R, can be used to apply hypothesis testing for the significance of the regression model. $\left|\overset{\wedge }{\beta_{i}}\right|\;$(i = 1,2,3…m), where m is the number of independent variables. In this paper, m = 3 denotes the three parameters, which are length-to-width, length-to-depth, and width-to-depth ratios of the pit. When the three variables’ values are relatively small, the stress concentration coefficient is considered not to be linearly regressed significantly with the three ratios, and we can assume the following:(5)$$H_{0}\overset{\wedge }{:\beta_{i}}=0\;(i\;=\;1,2,3\ldots m)$$

Statistical indicators are defined as follows:(6)$$\tilde{F}=\frac{\sum_{i=1}^{n}{\left({\overset{\wedge }{F}}_{i}-\bar{F}\right)}^{2}/m}{\sum_{i=1}^{n}{\left(F_{i}-{\overset{\wedge }{F}}_{i}\right)}^{2}/\left(n-m-1\right)}\sim F\left(m,n-m-1\right)$$

In Equation (6), $\bar{F}=\sum_{i=1}^{n}F_{i}/n$ and the statistical index $\tilde{F}$ obeys the F -distribution with degrees of freedom of (m,n−m−1). If $F>F_{1-\alpha }\left(m,n-m-1\right)$, the hypothesis test can be rejected, showing that a significant regression linear relationship exists among the stress concentration and the three parameters (the ratio of length to width, the ratio of length to depth, and the ratio of width to depth) within the confidence level of 1−α.

No matter the single linear regression or multiple linear regression, the significance is usually tested by the correlation coefficient. The solution method to the correlation coefficient is as follows:(7)$$R=\sqrt{\sum_{i=1}^{n}{\left({\overset{\wedge }{F}}_{i}-\bar{F}\right)}^{2}/\sum_{i=1}^{n}{\left(F_{i}-\bar{F}\right)}^{2}}$$

A larger R means better regularity and higher correlation. When R>0.8, the stress concentration factor can be considered to be highly linearly related to the three parameters (the ratio of length to width, the ratio of length to depth, and the ratio of width to depth) of the pits [11,12].

### 5.3. Significance Test of Regression Coefficient

Rejection of the hypothesis test for the regression model indicates that the regression coefficients ${\overset{\wedge }{\beta }}_{i}$ (i = 1,2,3…m) are not all close to zero, but does not exclude the possibility that some parameters are close to zero. When a certain parameter ${\overset{\wedge }{\beta }}_{i}$ is close to zero, the linear relationship between the parameter and the stress concentration factor is not significant, and the original hypothesis can be assumed:(8)$$H_{0}^{\left(i\right)}\overset{\wedge }{:\beta_{i}}=0$$

Statistical indicators are defined as follows:(9)$$t_{i}=\frac{{\overset{\wedge }{\beta }}_{i}/\sqrt{diag\left[{\left(\Delta^{T}\Delta \right)}^{-1}\right]}}{\sqrt{\sum_{i=1}^{n}{\left(F_{i}-{\overset{\wedge }{F}}_{i}\right)}^{2}/\left(n-m-1\right)}}\sim t\left(n-m-1\right)$$

Statistical index $t_{i}$ follows the t -distribution with degrees of freedom of (n−m−1). If $\left|t_{i}\right|>t_{a}\left(n-m-1\right)$, the assumption is rejected, indicating that significant differences exist between $\overset{\wedge }{\beta_{i}}$ and zero at the confidence level 1−α, which further illustrates that this parameter has a great impact on the stress concentration factor and needs to be retained in the matrix. In contrast, while $\left|t_{i}\right|<t_{a}\left(n-m-1\right)$, no significant differences exist between $\overset{\wedge }{\beta_{i}}$ and zero within the confidence level 1−α, and the effect of this parameter on the stress concentration factor should be ignored.

### 5.4. Prediction of Multiple Regression Model

Having tested the multiple regression model and coefficients, the regression relationship between the stress concentration coefficient and the pit length-width ratio, length-depth ratio, and depth-width ratio can be expressed as follows:(10)$$\overset{\wedge }{F}={\overset{\wedge }{\beta }}_{0}+{\overset{\wedge }{\beta }}_{1}\Delta_{1}+{\overset{\wedge }{\beta }}_{2}\Delta_{2}+\cdots {\overset{\wedge }{\beta }}_{m}\Delta_{m}$$

The test results and numerical simulation results are shown in Figure 17, where the relationship between the stress concentration factor and the length-width, length-depth, and width-depth ratios of the pits are drawn. These calculation results are substituted in Equation (1) to (10) for multidimensional linear regression analysis to analyze the influence of various parameters on the stress concentration coefficient.

The correlation coefficient R^2^ = 0.886 of the regression equation is obtained by multi-dimensional linear regression, and the basic parameters of the regression equation are shown below.

Table 4 illustrates that the test levels of parameters are all within the confidence interval. The correlation coefficient R^2^ = 0.886 of the regression equation shows that the fitting results are highly linearly correlated with the measured results, which indicates the feasibility of the regression results. The law between the stress concentration factor and the length–width, length–depth, and width–depth ratios of the pits is shown below:(11)$$SCF=F_{i}=-1.633+2.107a_{i}-0.476b_{i}+0.741c_{i}$$

## 6. Corrosion Fatigue Life

The fatigue life of a component can be generally classified into two stages: the formation life and the expansion life of cracks. The fatigue crack formation life often refers to the life beginning with the fatigue crack formation and ending with the inspectable length in engineering. Fatigue cracks are prone to occur where concentrated local stress and strains of components appear. Owing to the stress and strain concentration phenomenon, the component reaches the stage of plastic deformation, which facilitates the initiation and expansion of the crack. Analyzing the local stress and strain of the components to predict the fatigue crack formation life, this method is called the local stress–strain method, and the basic idea is as follows: the keys to determining the fatigue crack formation life of a component are the maximum local stress and strain in the local stress–strain concentration area. As long as the maximum local stress and strain are the same, so are the fatigue life of the components. With the fatigue life curve of the smooth material specimen, the fatigue crack formation life can be calculated corresponding to the maximum local stress and strain.

The fatigue life assessment of the structure by the Neuber′s method is conservative, which has been widely used in practical engineering. According to the basic view of the local stress–strain method, the strain fatigue life curve of a smooth material specimen is shown in Equation (12).
(12)$$\frac{\Delta \varepsilon }{2}=\frac{{\sigma^{\prime }}_{f}-\sigma_{0}}{E}{\left(2N_{i}\right)}^{b}+{\varepsilon^{\prime }}_{f}{\left(2N_{i}\right)}^{c}$$
where Δε is the stress range, $\sigma_{0}$ denotes the average stress, E means the elastic modulus, $N_{i}$ is the fatigue life of the material, ${\sigma^{\prime }}_{f}$ denotes the fatigue strength coefficient, and ${\varepsilon^{\prime }}_{f}$ means the fatigue ductility coefficient.

Δσ and Δε meet the following relationship:(13)$$\Delta \sigma \Delta \varepsilon =\frac{{\left(K_{f}\Delta s\right)}^{2}}{E}$$
where $K_{f}$
is the fatigue notch factor, ρ means the pit depth, a denotes the material parameter, SCF is the stress concentration factor, and Δs means the alternating load stress amplitude. As shown in Equation (14),
(14)$$K_{f}=1+\frac{SCF-1}{1+\sqrt{a/\rho }}$$

The relationship between the stress variation and stress amplitude is as follows:(15)$$\frac{\Delta \sigma^{2}}{4E}+\frac{\Delta \sigma }{1+n}{\left(\frac{\Delta \sigma }{2K_{e}}\right)}^{\frac{1}{n}}=K_{f}{}^{2}\frac{\Delta s^{2}}{4E}$$
where $n=\frac{b}{c}$, where b is the fatigue strength index and c denotes the fatigue ductility index, which satisfies the following relationship:(16)$$b=-\frac{1}{6}\mathrm{lg}\left[2(1+\frac{344}{\sigma_{b}})\right]$$
where $K_{e}$ is the cyclic strength coefficient; Δσ can be obtained by Formula (17); and the expression of the average stress can be obtained according to the relationship between the average stress $\sigma_{0}$ and the maximum stress $\sigma_{\max}$, expressed as follows:(17)$$\sigma_{0}=\frac{2\sigma_{\max}-\Delta \sigma }{2}$$
where Δε can be obtained by the simultaneous Equations (13)–(15). ${\sigma^{\prime }}_{f}$ and ${\varepsilon^{\prime }}_{f}$ satisfy the following expression:(18)$${\varepsilon^{\prime }}_{f}={\left(\frac{{\sigma^{\prime }}_{f}}{K_{e}}\right)}^{\frac{1}{n}}$$
(19)$$K_{e}=0.608\sigma_{b}{\left(0.002\right)}^{-n}$$

The values of the parameters during the entire operation are as follows:

The elastic modulus of the steel strand is equal to 2 × 10^11^; the tensile strength of the steel strand $\sigma_{b}$ is 1860 MPa; c is the fatigue ductility index (for ductile materials, take −0.6 [26]); ${\sigma^{\prime }}_{f}$ denotes the fatigue strength value determined by tests; and the values are 2260 MPa, 2050 MPa, and 1950 MPa when Δs is 100 MPa, 200 MPa, and 300 MPa, respectively.

The relationship between the stress concentration factor and the fatigue corrosion life of the steel strands can be obtained by substituting the results of the numerical simulation and testing them in the above formula. The corresponding experience curve drawn is shown in Figure 18. The ordinate $N_{i}$ denotes the corrosion fatigue life of the steel strands, and the abscissa SCF means the stress concentration factor of the pit. The figure depicts that when the stress concentration coefficient changes during 1–2, the fatigue life of the steel strands fluctuates greatly. When the stress concentration factor exceeds 2, the effect of the stress concentration factor on the fatigue life of the steel strands will not be significant, and the fatigue life will approach zero, which shows that expanding pits result in a faster derivation rate that may damage the steel strands instantaneously.

Composed of multiple strands of steel wire coupled with each other, the outstanding mechanical performance of steel strands results from the stranding and friction between the steel wires. Under the coupling of environmental and alternating loads, the corrosion fatigue life of each wire is different; when exposed to the external environment (high concentration of chloride) for a long time, the outermost steel wire of the steel strand will corrode faster, while the inner wire corrosion rate is slower owing to the low concentration of chloride ion inside. Accordingly, from the time the outer steel strand cracks until the damage, the inner steel wire can still provide the bearing capacity. However, once the outer steel wire is corroded and broken, the stranding and friction between the wires will gradually vanish, leading to the weak mechanical performance of the whole steel strands compared with the uncorroded steel strands. For the sake of safety, the mechanical properties of the steel strand are considered to have failed simultaneous in engineering.

## 7. Conclusions

For this paper’s study, an indoor physical experiment for steel strands specimens exposed to the salt spray environment is carried out. The specimens are tested under the action of different load amplitudes. The corrosion of steel strand is observed at various exposure times, and we find that deeper and sharper pits result in greater pitting-local stress and a shorter fatigue life; namely, pits play a significant role in the life of the steel strands. In order to predict the corrosion life, a prediction model consisting of the stress concentration factors is set up to simulate the corrosion life of the steel strands considering the coupling effects of the fatigue caused by the cyclic loading and corrosion phenomenon.

The conclusions are shown as follows:(1)An indoor experiment for steel strands specimens under different load amplitudes exposed to the salt spray environment is carried out, and the pit depths at various exposure times are tested. The evolution laws of the pits are obtained based on the neural network method, and the conclusions indicate that the proposed prediction model is validated by comparing the pit depth obtained from the prediction model to the experimental measurements. Additionally, the corrosion rate will decrease as time flows.(2)The corrosion life of steel strands is mainly influenced by the sizes of the pits. Deeper and sharper pits result in greater pitting-local stress and a lower fatigue life. An empirical formula consisting with the ratio of the length to width, the ratio of the length to the depth, and the ratio of the width to the depth of the pits is obtained to determine the stress concentration factor based on the multi-dimensional linear regression method. The results show that, with a larger length–width ratio and width–depth ratio, the stress concentration factor will increase. On the contrary, the length to depth ratio is inversely proportional to the stress concentration factors.(3)The fatigue notch factor of components can be deduced by the stress concentration factor, and the life of the steel strands can be shortened by both of them. The conclusions indicate that the fatigue life of the steel strand fluctuate greatly when the stress concentration factor changes during 1–2, whereas when the stress concentration factor exceeds 2, the effect of the stress concentration factor on the fatigue life of the steel strands will not be significant, and the fatigue life will approach zero, which shows that expanding pits lead to a faster derivation rate, which may damage the steel strands instantaneously. The findings are expected to be useful in realistically predicting the durability of the wire structures. Further, the fatigue curve theory should be further investigated and developed.

## Figures and Tables

**Figure 1 materials-13-00736-f001:**
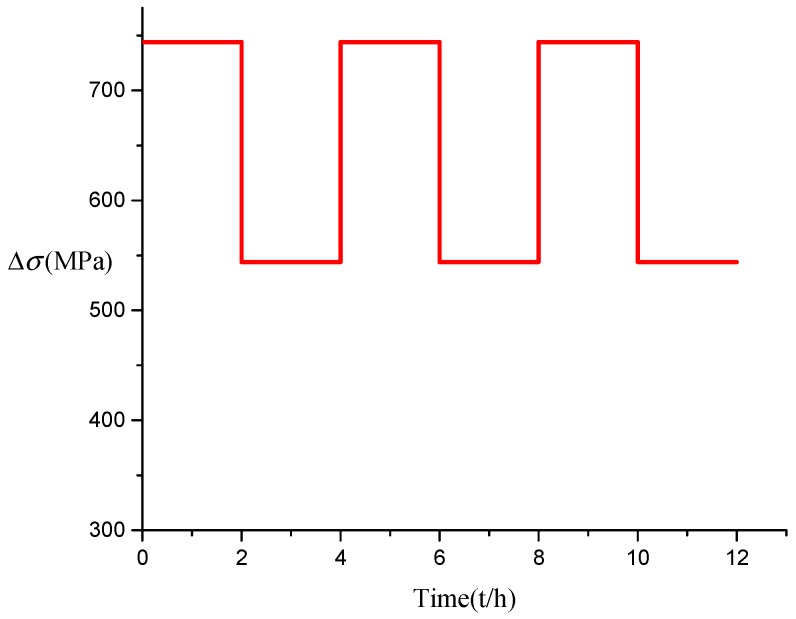
Loading schematic diagram.

**Figure 2 materials-13-00736-f002:**
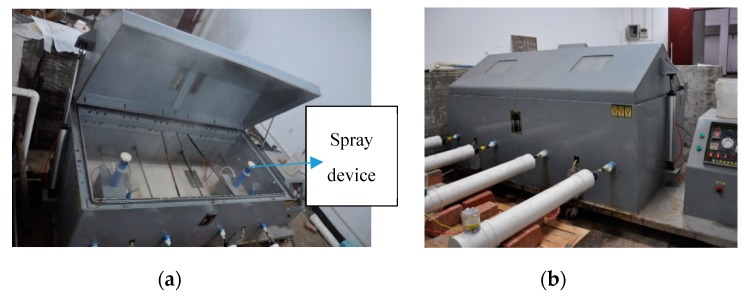
(**a,b**) Spray device.

**Figure 3 materials-13-00736-f003:**
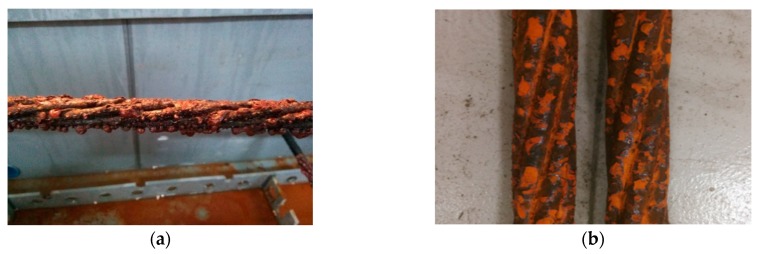
(**a,b**) Corrosion results of the steel strand.

**Figure 4 materials-13-00736-f004:**
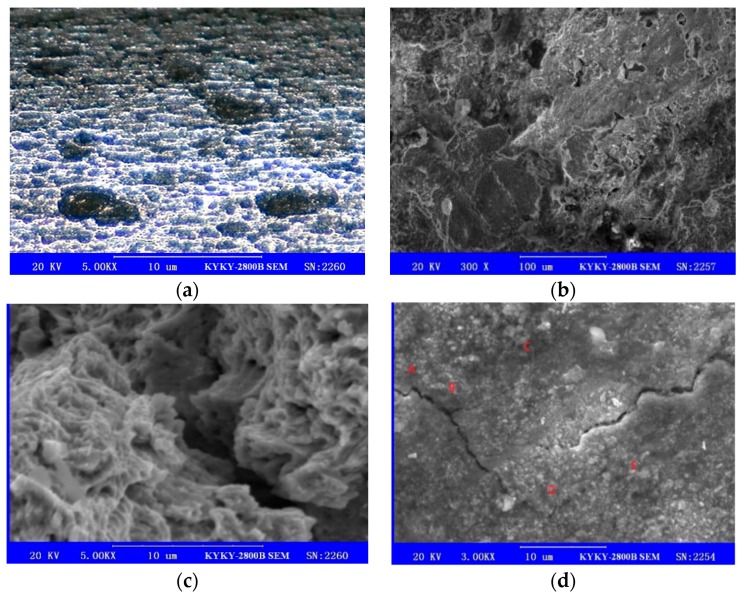
(**a**–**d**) Pit morphology.

**Figure 5 materials-13-00736-f005:**
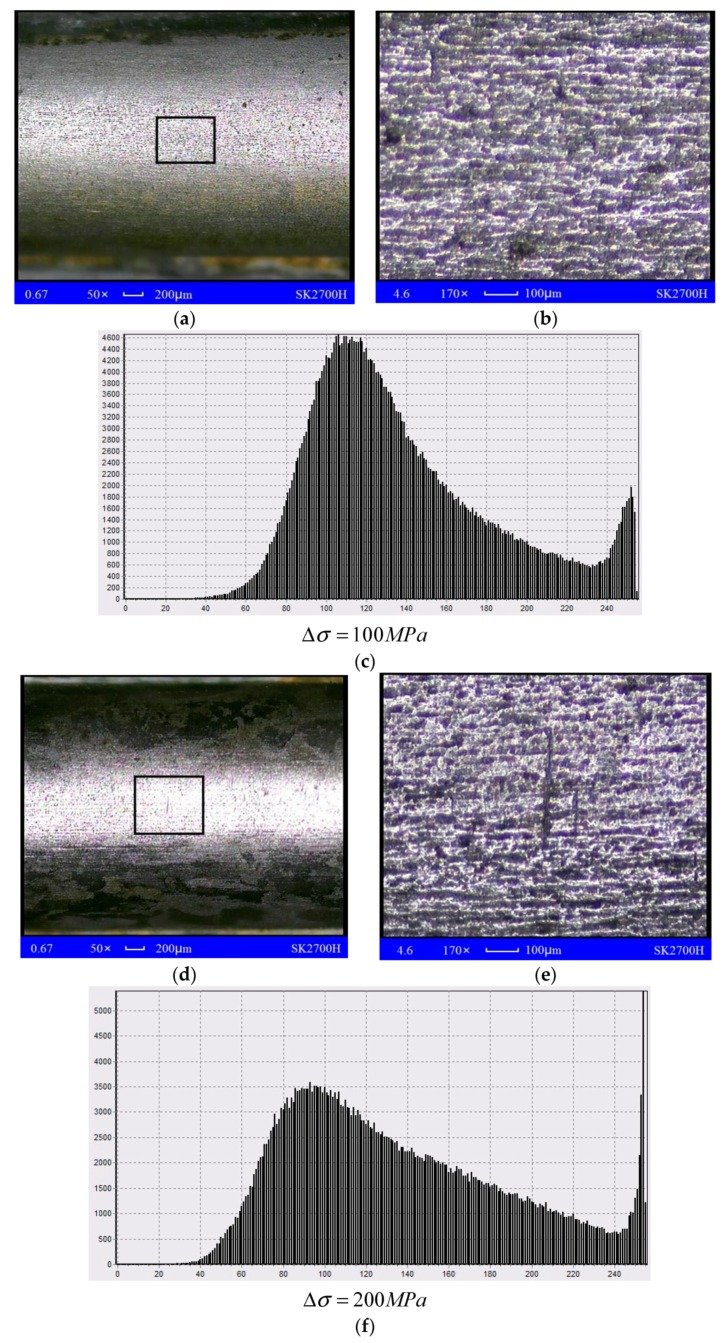
Test results (**a**,**b**,**d**,**e**,**g**,**h**); Local corrosion histogram of steel strand (**c**,**f**,**i**).

**Figure 6 materials-13-00736-f006:**
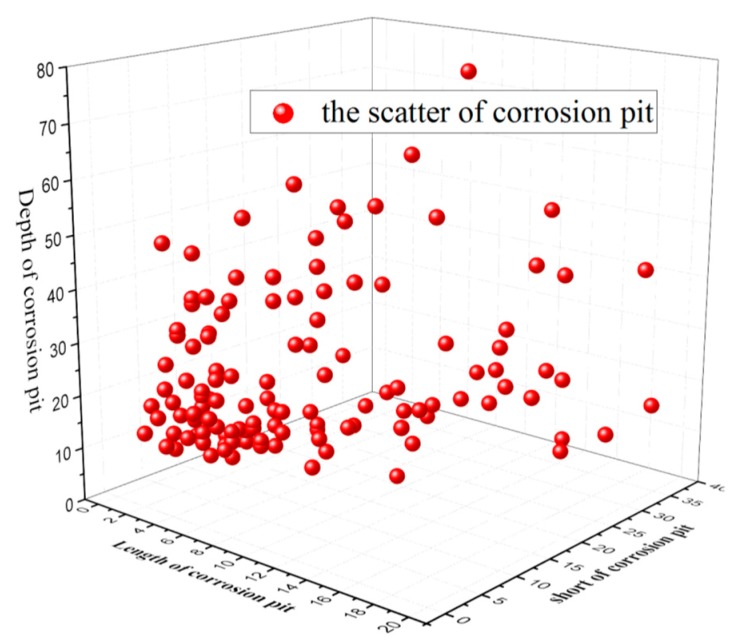
The size of the corrosion pit.

**Figure 7 materials-13-00736-f007:**
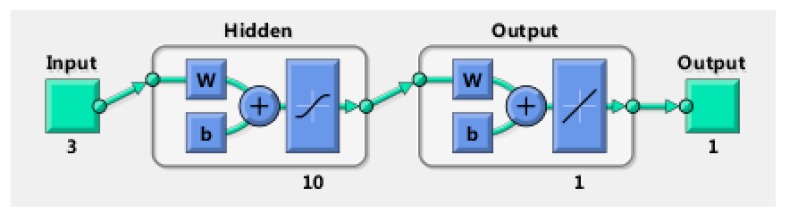
Structure diagram of the neural network.

**Figure 8 materials-13-00736-f008:**
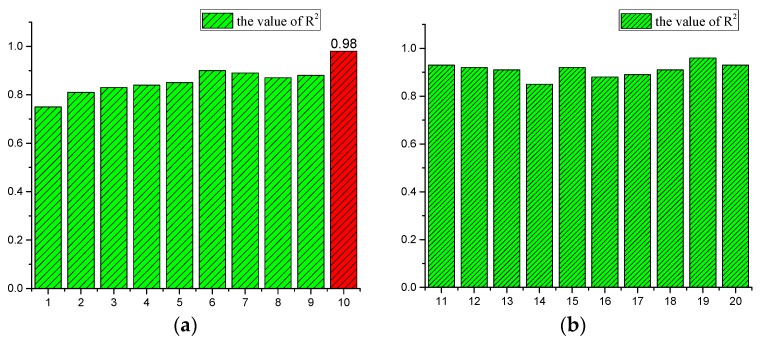
(**a**,**b**) Fitting degree of each level.

**Figure 9 materials-13-00736-f009:**
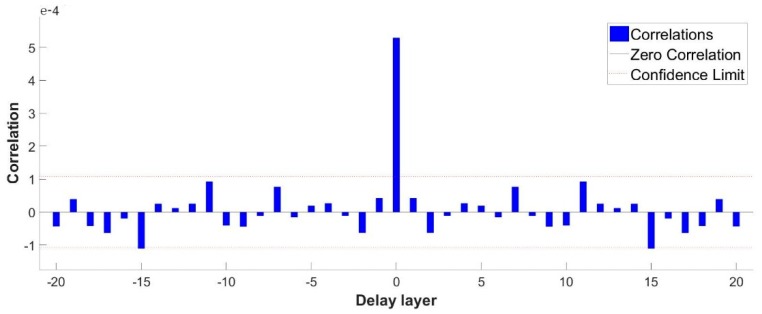
Autocorrelation error.

**Figure 10 materials-13-00736-f010:**
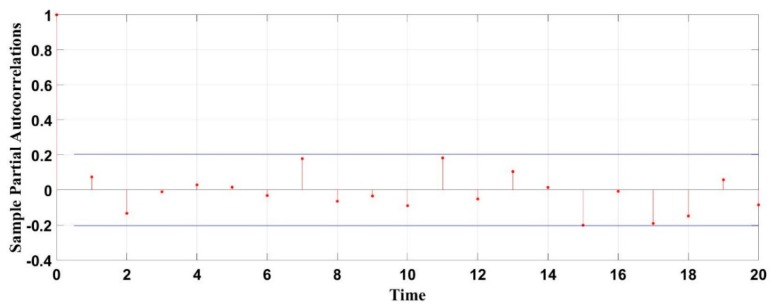
Error diagram of target value and output value.

**Figure 11 materials-13-00736-f011:**
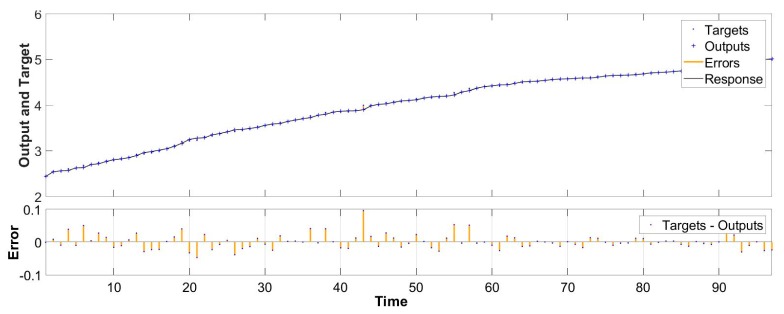
Comparison chart of prediction results.

**Figure 12 materials-13-00736-f012:**
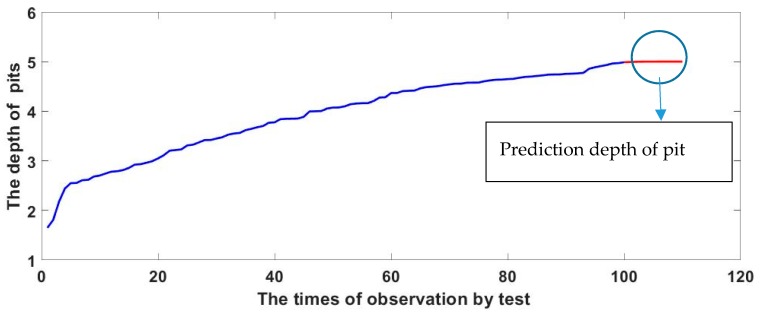
Pit prediction model.

**Figure 13 materials-13-00736-f013:**
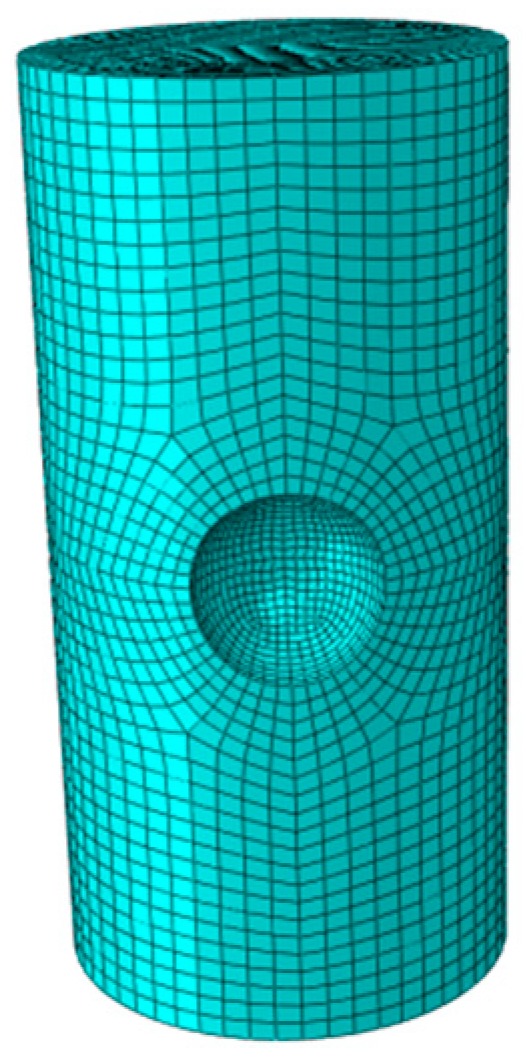
Steel strand grid.

**Figure 14 materials-13-00736-f014:**
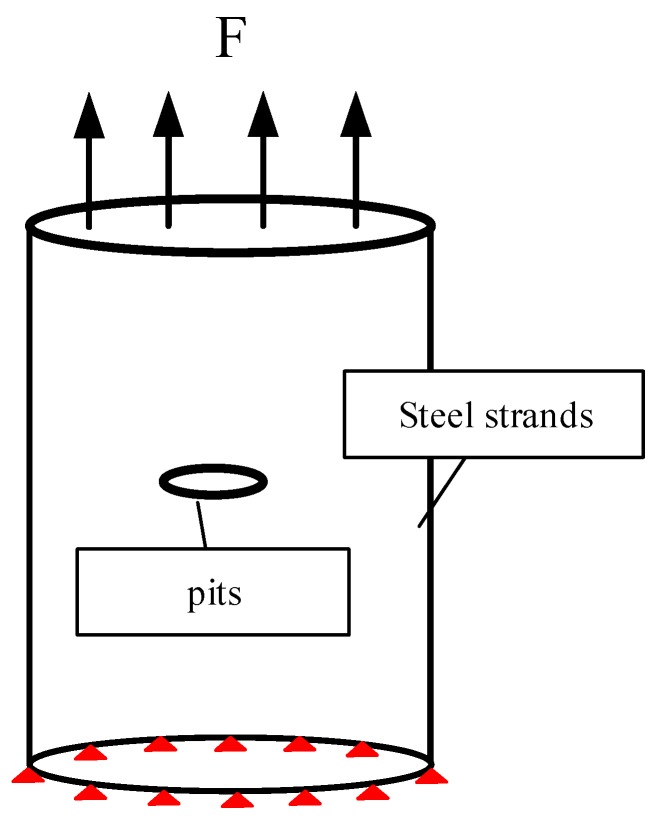
Schematic diagram of model loading.

**Figure 15 materials-13-00736-f015:**
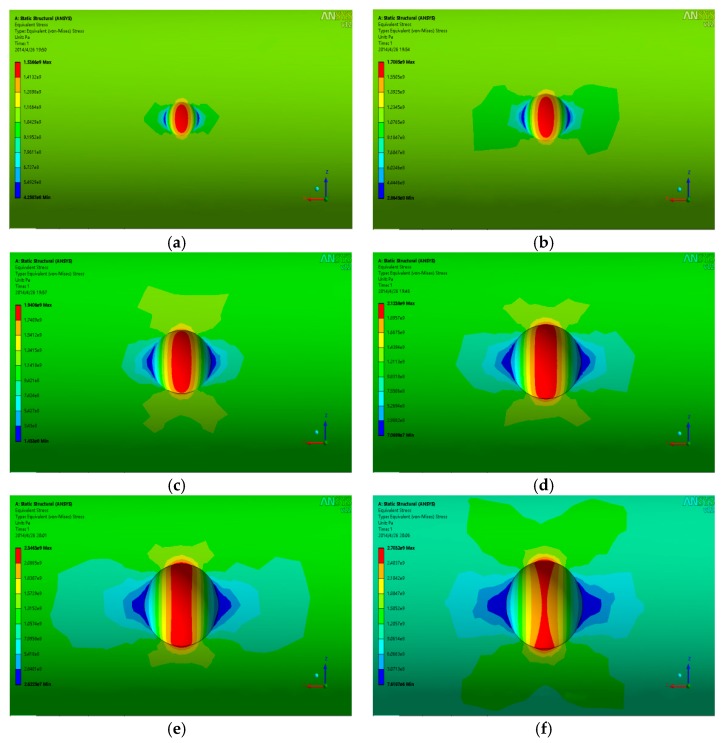
Pit stress diagram (considering depth only). Pit depth (**a**) d = 0.1 mm; (**b**) d = 0.2 mm; (**c**) d = 0.4 mm; (**d**) d = 0.6 mm; (**e**) d = 0.8 mm; (**f**) d = 1 mm.

**Figure 16 materials-13-00736-f016:**
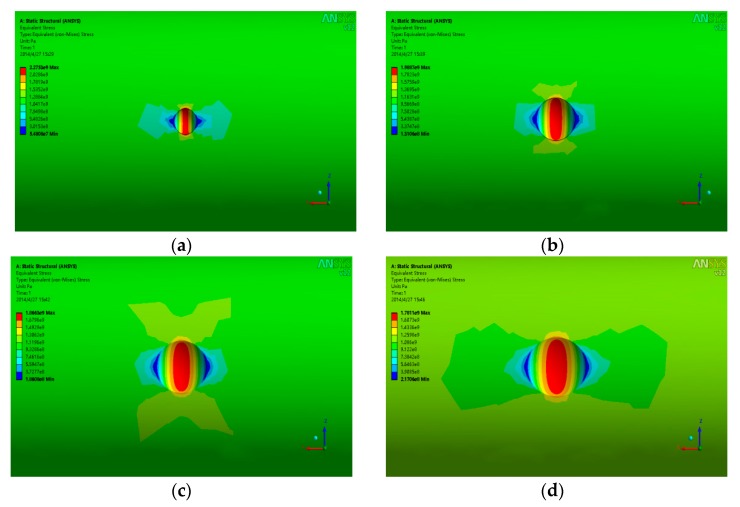
Three-dimensional simulation results of pit stress. (**a**) w/d = 2; (**b**) w/d = 3.3; (**c**) L/w = 1; (**d**) L/w = 1.5.

**Figure 17 materials-13-00736-f017:**
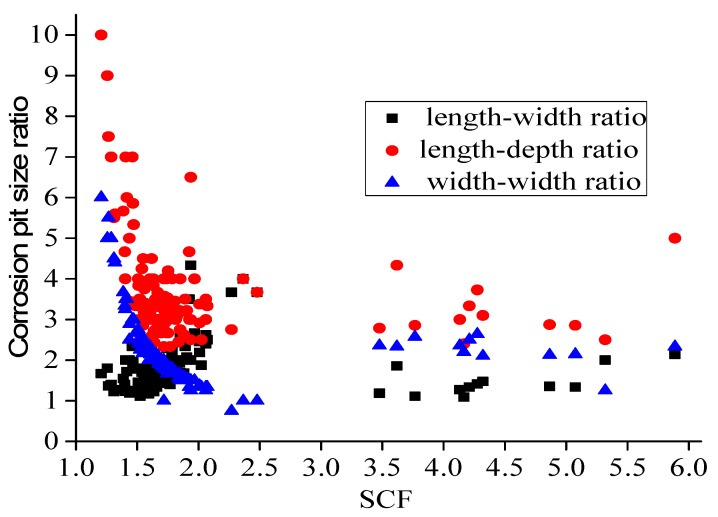
The relation between stress concentration factor and pitting ratio. SCF, stress concentration factor.

**Figure 18 materials-13-00736-f018:**
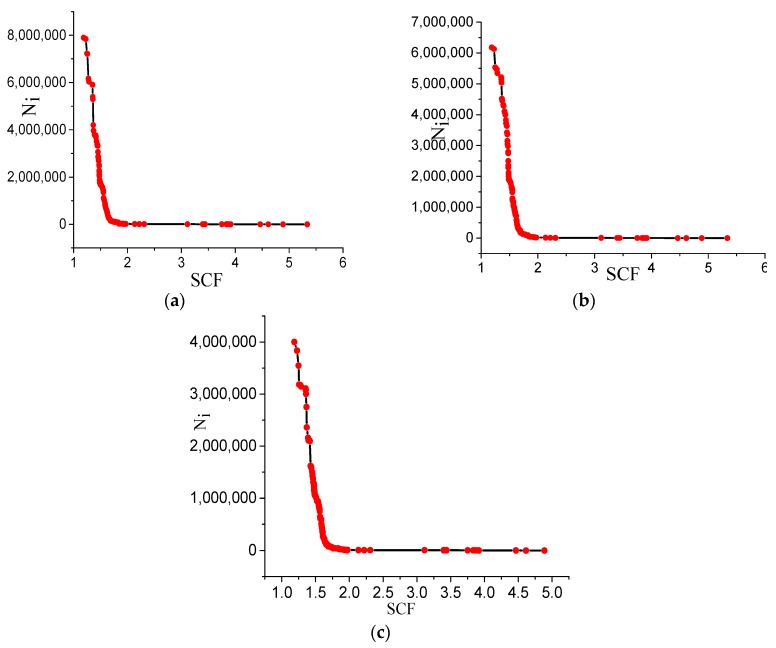
Fatigue life curve of steel strand. (**a**) ΔS = 100 MPa; (**b**) ΔS = 200MPa; (**c**) ΔS = 300 MPa.

**Table 1 materials-13-00736-t001:** Test equipment.

Device Name	Type	Technical Parameter	Quantity (Unit)
Salt fog test box	YC-200	Saline spray deposition 250 ml/m2/h	1
Universal testing machine	WAW-1000	Maximum load 1000 kN	1
Air compressor	VB-0.2/8	Power 2.2 KW	1
Digital scanning electronic Instrument	KYKY-2008B	-	1
Electronic balance	SL500ZN	Accuracy 0.01 g	1
Ultra-high voltage electric oil pump	DSS	Power 0.75 KW	1
Acidimeter	PHS-3C	-	1

**Table 2 materials-13-00736-t002:** Chemical composition of the salt solution.

Chemical Component	NaCl	H_2_O	CuCl_2_·2H_2_O	CH_3_COOH
Content (5000 mL)	250 g	4718.7 mL	1.3 g	30 mL
Percent	5%	94.37%	0.03%	0.60%

**Table 3 materials-13-00736-t003:** Grayscale histogram data under the loading condition.

Loading Status	Hue Level (Peak)	Standard Deviation	Pit Frequency (Tone ≤ 110)	Total Frequency	Pit Density
Δσ=200 MPa	93 (3595)	1121.2	156,777	398,034	39.4%
Δσ=300 MPa	73 (4150)	1091.9	218,080	398,034	55.8%
Δσ=100 MPa	96 (4671)	1440.4	126,541	398,034	32.1%

**Table 4 materials-13-00736-t004:** Parameter values of the multidimensional linear regression equation.

Regression Parameter	Parameter Value	Value (t)	Inspection Level
*β* _0_	−1.633	−7.376	1.6 ×10^−11^
*β* _1_	2.107	17.779	0
*β* _2_	−0.476	−6.561	1.1 ×10^−9^
*β* _3_	0.741	6.214	6.3 ×10^−9^

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
