# Peer review of "Numerical and Experimental Study on the Steel Strands under the Coupling Effect of a Salt Spray Environment and Cyclic Loads"

_materials, 2020, doi:10.3390/ma13030736_

Round 1
Reviewer 1 Report
This paper presents an interesting, well-conceived and relevant study on the corrosion of steel wires used for cables in infrastructure applications.
The study presents experimental findings on accelerated corrosion testing and characterization of steel cable strands - and uses those results to inform analysis and modeling assessment of the impact of corrosion events on mechanical performance, load-bearing capacity and fatigue performance. The authors are commended for putting together a complete study based on their novel testing and findings.
However, before publication, there are a few issues that must (in this reviewers opinion) be addressed:
-there are many instances of the English language used throughout the article that make the article difficult to interpret in some instances. It should be proofread and edited thoroughly for English usage. Given more time I could be available to provide line by line edit suggestions. For example, the word "tremendous" is used frequently but does not accurately describe behaviors being described and the word "obvious" should be avoided (to avoid insulting readers) and instead what is obvious should be accurately described to avoid confusion.
-Section 2: Some experimental details are missing and should be included for example: the wire composition, surface prep (washed? degreased?) and thickness/composition of the zinc coating. Also the loading frequency of the stress amplitude is unclear. Also how many wires are in each strand?
-Figure 1: inset text should be "spray device"
-Figure 5: drop lines from the data points should be added otherwise it is impossible to see the x,y,z position of the data points.
-Figure 11: Chinese characters in figure inset.
-Sections 3.2 and 5: and similar contain quite a bit of background type information and should be considered to move to Supplementary Info, to avoid excessive length and diluted focus of the paper.
-Can the authors comment of the influence of erosion-corrosion effects concerning the small amplitude motion between wires in contact within a strand?
-References: when possible it would be good to include english language references or access to english translations.
Reviewer 2 Report
In my opinion, the problem is much more complex. However, it is important to show the effects of corrosion and fatigue sensitivity to corrosion pits.
A weak element of the work is the numerical FEM model that models the pitting in an idealized way because the SCF parameter depends on the results from the FEM model. fter all, the level of stress concentration may largely depend on the shape of the pitting.
Fig. 13 could be better described. Are the stresses shown read from the outer surface or deeper? Formally, it would be not bad to show loading direction.
General comment: The work shows the sensitivity of high-strength steels to corrosion and takes a step forward in investigating this problem.
